# On the Heterogeneity of Deformation and Fracture in Bimetallic Specimens of the C11000-Inconel 625 System

**DOI:** 10.3390/ma18235450

**Published:** 2025-12-03

**Authors:** Kseniya Osipovich, Vyacheslav Semenchuk, Andrey Chumaevskii, Alexander M. Korsunsky, Yuri Kushnarev, Evgeny Moskvichev, Alihan Amirov, Denis Gurianov, Sergei Tarasov, Evgeny Kolubaev

**Affiliations:** 1Institute of Strength Physics and Materials Science, Siberian Branch of Russian Academy of Sciences, 634055 Tomsk, Russia; svm_70@ispms.ru (V.S.); tch7av@gmail.com (A.C.); desa-93@mail.ru (D.G.);; 2Hierarchically Structured Materials (HSM) Laboratory, Center for Digital Engineering, Skolkovo Institute of Science and Technology, 30 Bolshoi boulevard, 121205 Moscow, Russia; 3Center for AeroSpace Materials & Technologies (CASM&T), Advanced Engineering School, Moscow Aviation Institute, 4 Volokolamskoe sh., 125993 Moscow, Russia

**Keywords:** wire-feed electron beam additive technology, 3D printing, bimetal, pure copper, nickel-based superalloy, macrostructure, mechanical properties, ultimate strength, fatigue strength, impact toughness

## Abstract

In this work, bimetallic specimens of the copper C11000-Inconel 625 system were fabricated using multi-wire electron beam additive technology. Three different sequences of component deposition were employed to produce the bimetallic specimens for investigation: Type A—nickel and pure copper were deposited side by side in parallel; Type B—layers of nickel-based superalloy were printed first, followed by the deposition of copper on top; Type C—copper layers were printed first, with nickel-based superalloy subsequently deposited on top. The influence of additive manufacturing conditions and sequence on the microstructure, static and fatigue strength, and impact toughness of the test pieces was studied. The results indicate the formation of a complex anisotropic structure in bimetals of various types during printing, driven by directional heat dissipation toward the substrate. The microstructure comprising large primary grains or dendrites elongated along the heat flow direction leads to significant differences in material properties along the printing (scanning) direction, the build (growth) direction, and at intermediate angles. Studies of the copper C11000-Inconel 625 bimetallic samples have shown that the interface between components does not exhibit inherent weakness compared to the base materials: pure copper or nickel superalloy. Tensile testing consistently reveals that fracture occurs by the adhesive mechanism in the weaker constituent, rather than at the interface.

## 1. Introduction

In recent years, there has been growing interest in combining dissimilar materials to create hybrid structures with complex properties [1]. The integration of dissimilar metals enables the full exploitation of the individual advantages of each component and represents an effective strategy for developing high-performance hybrid systems. Bimetallic materials allow for the combination of the advantages of different materials in terms of mechanical and/or physical properties, and, thus, for obtaining a synergetic effect [2]. Owing to their high wear and corrosion resistance, thermal stability, and creep resistance, the Ni-Cu alloy combination has attracted considerable attention in materials science and engineering [3]. Copper and its alloys constitute an important class of engineering materials, widely employed due to their exceptional thermal and electrical conductivity, ductility, corrosion resistance, and other favorable properties. These characteristics render them indispensable in demanding sectors such as aerospace, transportation, and military applications [4]. Cooling systems for heavy-duty equipment, for instance, are typically fabricated from copper-based alloys. Copper exhibits high thermal conductivity (~400 W/m·K), making it particularly suitable for applications requiring efficient heat transfer or dissipation [5]. Nickel-based superalloys, in turn, offer excellent corrosion resistance, good mechanical properties, and significant strength retention at elevated temperatures [6,7,8]. These properties make nickel-based superalloys suitable for use as components in high-temperature regions where high thermal strength is required in the aerospace industry [9,10,11]. For example, a combustion chamber must withstand extreme conditions, including high pressure, wide temperature fluctuations, intensive vibration, and corrosive environments [12,13], necessitating stringent requirements for both material performance and structural reliability. Consequently, copper-based alloys are commonly selected for the inner wall due to their superior thermal conductivity, good erosion resistance at elevated temperatures, and good machinability [14,15], while nickel alloys are often used for the outer wall owing to their high strength, excellent weldability, and widespread use in high-precision electroforming processes [16,17].

Thus, nickel-based superalloys and copper-based alloys are well-suited as base materials for fabricating highly efficient joints in practical engineering applications. Given the extensive use of various copper-based alloys, there is a clear need to fabricate joints in dissimilar combinations characterized by mismatched coefficients of thermal expansion. However, due to the large difference in physical properties, challenging problems may arise, such as copper penetration into grain boundaries and thermal cracking [18], which is potentially hazardous in terms of jeopardizing the corrosion resistance of the joint and reducing its safety. In recent years, researchers have explored various welding techniques such as friction stir welding [19], explosion welding [20], and electromagnetic pulse welding [21] for joining such dissimilar material pairs. Nevertheless, these methods often impose strict geometric constraints on joint design, limiting their applicability in complex or large-scale manufacturing scenarios. Soldering is an alternative joining approach for dissimilar materials, including copper-based components [22,23]. Advances in high-performance manufacturing of metals and alloys have yielded increasingly sophisticated and cost-effective production strategies [24,25,26,27]. In particular, wire-based additive manufacturing offers significant advantages over conventional techniques by reducing both production time and cost. Among the primary wire-fed 3D printing technologies are wire arc additive manufacturing (WAAM) and wire electron beam additive manufacturing [28,29,30]. Wire and electron beam additive manufacturing (WEBAM) is especially well-suited for producing components with stable microstructures, high ductility, and favorable fatigue performance [31]. Significant progress has recently been achieved in the WEBAM of titanium alloy components and bimetallic structures based on nickel and copper alloys [32,33,34]. WEBAM is an indispensable technology in the production of large-sized products made of copper or low-alloy bronzes and bimetallic elements based on them [35,36,37]. This situation determines the considerable interest in this technology on the part of the aviation and space sectors of industry. The production of bimetallic copper–nickel chambers and guide vanes, as well as other components for aerospace engines, and many other multi-component elements for industrial applications using WEBAM is becoming significantly more technologically feasible and economically advantageous.

Although the mechanical and microstructural characteristics of the traditionally made CuNi alloy are extensively documented [38,39,40,41,42,43,44], research on analogous AM versions, particularly concerning the connection between microstructure and mechanical properties, is still scarce. The possibility of producing Cu–Ni alloys through laser metal deposition using mixed powder feedstocks with different Cu/Ni weight ratios was explored [45]. Tensile tests indicated that raising the copper content in nickel enhances strength through solid solution strengthening and better ductility. Creating a complete composite gradient Cu–Ni alloy through in situ alloying via microlaser powder bed melting enables a thorough examination of how the microstructure and properties of the alloy vary with composition [46]. The compacted composite gradient Cu–Ni alloy demonstrated ongoing microstructural transformations and mechanical, electrical, and thermal properties that vary with composition, facilitating seamless property variations devoid of interfaces and providing design principles for multifunctional intricate components. Bypass current plasma arc welding (BC-PAW) was employed to create thin-walled Cu–Ni gradient structures [47]. Baraz et al. [48] applied Cu81-Ni19 layers onto an annealed substrate created via a traditional technique and discovered that the hardness of the layer formed by the PBF-LB method was 19% greater than that of the annealed base material. They ascribed this increase in hardness exclusively to the quasi-amorphous characteristics of the deposited layer but provided no further clarification or corroborating evidence. Maleta et al. [49] studied the microstructure and mechanical characteristics of WAAM 90/10 CuNi and found that the WAAM material exhibited greater toughness than its cast counterparts. Williams et al. [50] recently assessed the correlations among process, structure, and properties in laser-assisted powder energy deposition (LP-DED) of 65/35 CuNi, discovering that essential first-layer connections between processing parameters, melt track morphology, defect development, and microstructure evolution serve as the foundation for optimizing the deposition of subsequent layers and consequently enhancing thermoelectric performance.

Nevertheless, several unresolved issues of both technological and scientific importance persist regarding the creation of titanium alloys with enhanced mechanical characteristics, the production of bimetallic components featuring various gradient zone designs, and the determination of how stress states impact the mechanical properties of the finished parts or assemblies.

Thus, this article aims to examine the variability in deformation behavior and fracture within bimetallic specimens of the copper C11000-Inconel 625 system produced via WEBAM.

## 2. Materials and Methods

Bimetallic specimens were produced using the experimental laboratory equipment, a 3D electron beam wire-feed printer at the ISPMS SB RAS. The specimens were produced using 1.6 mm diameter wires of pure copper C11000 and nickel-based superalloy Inconel 625. The chemical compositions of the as-received wires are provided in Table 1.

Three different sequences of component deposition were tested during printing (Figure 1). Type A bimetallic specimens were produced by depositing the nickel-based superalloy and pure copper side by side in a parallel arrangement (Figure 1a). To minimize intermixing at the interface, a portion of the nickel-based superalloy component was pre-deposited before copper deposition began. Specifically, an initial ~8 mm high nickel component was built directly on the substrate (Stage 1). This was followed by the deposition of two 1 mm copper layers (Stage 2). Subsequently, two further 1 mm nickel layers were added. Stages 1 and 2 were repeated alternately until the required height was reached. For the Type B bimetallic specimens, a nickel-based superalloy wall of ~50 mm in height was printed first, and then layers of pure copper with a height of ~50 mm were deposited on top of it (Figure 1b). When manufacturing the Type C bimetallic specimens with similar dimensions, the copper part was printed first, after which layers of nickel-based superalloy were deposited onto it (Figure 1c).

The first stage of nickel-based superalloy printing was performed with a layer-by-layer reduction in heat input by decreasing the beam current. Then the copper component of the part was printed with an increase in heat input according to the data presented in Table 2. The remainder of the nickel alloy part of the component was printed at a constant beam current of 60 mA. The copper part of the Type B bimetallic component was printed at a constant beam current of 55 mA, and a pause of 15–25 s was maintained between passes to prevent overheating and excessive material spreading. Other parameters for printing with nickel superalloy and pure copper are shown in Table 2. Printing was performed on a 5 mm thick AISI 321 stainless steel substrate at an operating pressure in the vacuum chamber in the range from 4 × 10^−3^ Pa to 7 × 10^−3^ Pa.

The test pieces were cut from the bimetallic specimens using a DK7750 electrical discharge machine (Stankoinstrument, Ekaterinburg, Russia) to study their microstructure and mechanical properties (Figure 2).

Test pieces were cut from the built plates to determine the strength characteristics in different areas of the component in the direction of growth, the direction of printing, and at an angle of 45°. For impact toughness testing, the test pieces were cut oriented in the growth direction and the printing direction for Type A bimetallic specimens. The test pieces for impact toughness testing were cut from Type B and Type C bimetallic specimens only in the growth direction. Separate test pieces were cut to determine the properties of the nickel and copper components and the structural gradient zone. The test pieces were also cut for tensile and low-cycle fatigue tests along the printing direction and the growth direction. For microstructure studies and X-ray structural analysis, test pieces were cut from various locations within the bimetal built plate and at the interface. For macrostructure studies, differently oriented test pieces (coupons) were extracted. Finally, the second and third types of test pieces were extracted (see below), with the position of the V-shaped cutout aligned with the biomaterial interface when cutting out the test pieces for impact toughness testing.

For structural studies, sections were ground using sandpaper with a grit size of 180 to 2000, followed by polishing with diamond paste. At the final stage, polishing was performed using an emulsion with silicon carbide particles of 0.05 μm, followed by sequential etching in selected reagents for pure copper (10 mL HCl + 1 g FeCl_3_ + 20 mL H_2_O) and for nickel-based superalloy (8 g CuSO_4_ + 40 mL C_2_H_5_OH + 40 mL HCl at a temperature of 80 °C) with intermediate repolishing. Macroscopic images of the bimetallic specimens were taken with a Pentax K-3 camera (Tokyo, Japan) with a lens focal length of 100 mm. Structural studies of the test pieces were carried out using an optical microscope Altami MET-1C (Saint-Petersburg, Russia), laser scanning microscopy Olympus LEXT 4100 (Tokyo, Japan), scanning electron microscopy (SEM EDS) on a Zeiss LEO EVO 50 microscope (Oberkochen, Germany), and transmission microscopy on a JEOL JEM-2100 (Tokyo, Japan). To obtain fractography images of the fracture surfaces of the test pieces, a scanning electron microscope with a Schottky cathode Tescan MIRA 3 LMU (TESCAN ORSAY HOLDING, Brno, Czech Republic) equipped with an energy-dispersive X-ray spectrometer Oxford Instruments Ultim Max 40 (Oxford Instruments, High Wycombe, UK) was used. The structure analysis was carried out with the following parameters: accelerating voltage, 20 kV; beam current, 1.6 nA. The analysis results were processed in AZtec v.4.2 licensed software (Oxford Instruments, High Wycombe, UK). The test pieces for SEM analysis were cut perpendicular to the printing plane and were polished with diamond suspension down to a 1 µm particle size, followed by colloidal silica polishing. To evaluate the changes in the phase composition, X-ray patterns were collected on a DRON-8N diffractometer (Burevestnik, Saint Petersburg, Russia). The angular range 2θ was set at 30–120°, the scanning step was 0.6°, and the exposure time was 5 s. Quantitative evaluation of crystal lattice microstrains was performed according to the Scherrer formula, assuming that all broadening of X-ray lines was caused by changes in Type B (grain level average) residual microstresses. The magnitude of Type B residual stresses was determined based on the simple Hooke’s law approximation σ = *E* ε, where ε is the microstrain values obtained from X-ray structural analysis (Scherrer broadening) and *E* is Young’s modulus.

Mechanical tests for the determination of static strength were performed on a universal testing machine UTS-110M (Test Systems, Ivanovo, Russia) at a crosshead movement speed of 1 mm/min. Fatigue strength tests were performed in a BISS-UT-04-100 testing machine (BISS, Bangalore, India) with a test base of 1,000,000 cycles, cycle stress amplitude of 0.3–0.7 of the ultimate tensile strength, and a frequency of 20 Hz. Tests on bimetallic specimens were performed in elastic and elastic–plastic modes. The minimum stress value in the cycle was selected at the initial stage of elastic deformation. Impact toughness was determined using an Instron 450 MTX pendulum tester (Instron, High Wycombe, UK). Test pieces contained a V-shaped concentrator of standard shape at the opposite side to the pendulum striker impact. Material microhardness was measured using a TVM 5215 A microhardness Vickers tester (Tochpribor, Saint Petersbourg, Russia) was used to determine local hardness values at different locations within the test pieces (upper, central, and lower parts). At each location, a set of three measurements was performed with a step of 0.1 mm, an exposure time of 10 s, and a load of 1 N.

## 3. Results and Discussion

### 3.1. Macrostructure of C11000-Inconel 625 Bimetallic Specimens of Types A, B, and C

Bimetallic specimens were fabricated using three different routes (Type A, B, and C) from two materials with distinct properties in order to aid in the selection of the most favorable fabrication protocol to obtain optimal performance of the final parts. Structure and manufacture-specific design of assemblies from heterogeneous material parts requires the knowledge of the physical and mechanical properties of additively manufactured materials to unlock their full potential and result in defect-free products with high structural integrity [51].

In the case of bimetallic components, a key objective is to avoid the formation of defects at the interface between the dissimilar materials and to verify the joint strength. Manufacturing defect-free multi-material pieces obtained from alternate feed wires requires control over the thermal conditions and ensuring that whilst the wire of material one melts fully, material two does not spread, causing defects and distortion of the part geometry.

This requires accounting for the physical and mechanical properties of both materials and calculating heat input values for each type of bimetallic specimen (Type A, B, and C) to minimize the remelting of previously deposited layers, spreading or incomplete melting of the wire fed into the melt pool, and control over the surface roughness. With optimal parameters (Table 2), the energy is evenly distributed throughout the volume, ensuring stable melting of dissimilar materials and the production of parts with the desired geometry. Images of the obtained bimetallic specimens are shown in Figure 3. It is noted that the edges reveal areas of unmelted material that appear in the form of needle-like wire remnants.

Figure 4a illustrates that the obtained bimetallic specimens of copper C11000-Inconel 625 systems of Type A consisted of two single-hatch walls (i.e., one additively formed layer thick) that were printed alternately parallel to each other with a continuous boundary. The macro images of the cross-section reveal the presence of irregular pores in the copper part in the upper segment and near the interface between copper and nickel superalloy. These defects are isolated, and their volumetric fraction does not exceed 0.2%.

Figure 4b shows that the macrostructure of the bimetallic specimens of copper C11000-Inconel 625 of Type B, with the nickel superalloy material IN625 first deposited onto the substrate and then over-printed with copper afterwards, exhibits a stable macrostructure. In the nickel superalloy part, one may observe characteristic interlayer boundaries typical of this fabrication regime. The thickness of these layers is approximately 400 µm. Due to differing etching rates of the materials used, it is difficult to deduce the structure in this area from the macrostructural image.

Figure 4c illustrates the bimetallic specimens of copper C11000-Inconel 625 systems of Type C, in which pure copper is first deposited onto the substrate and then is over-printed with nickel superalloy. Samples demonstrate a fairly uniform structure in both components. The copper part of the sample, similar to Type B specimens, is characterized by a grain structure in which three segments with different structural element sizes can be distinguished.

### 3.2. Structural and Phase State Analysis of Bimetallic Specimens of Copper C11000-Inconel 625 Types A, B, and C

The copper component of Type A specimens exhibits a grain structure characterized by the formation of large grains that emerge from preferential growth in the direction of heat flow (Figure 5a,b). The grain orientation in the copper part depends on the heat removal to the substrate and to the adjacent nickel component. For this latter reason, the grains are significantly deviated from the vertical direction. Near the interface with the nickel part, equiaxed grains are formed with an average size of 20 µm. Further away from the interface, large, elongated grains develop with the initial widths and lengths of 125 µm and 415 µm, respectively (Figure 5b).

The interface from the nickel part to the pure copper part of the specimen is sharp, with no defects such as cracks or lack of fusion detected, confirming the formation of a strong bond between the materials (Figure 5c,f). Since the interface is sharp, significant mixing of dissimilar materials does not appear to occur during its formation. The interface thickness does not exceed 5 µm. The nickel alloy part of the Type A specimen exhibits significant “over-etching” due to the different etching rates of the materials used, as well as complex conditions of 3D printing near the interface between the two materials. This complicates microstructure determination in the material macrostructural image near the interface (Figure 4). During the deposition of nickel layers, a dendritic structure forms. Microstructure images (Figure 5d,g) show that a complex heterogeneous dendritic structure develops in the nickel part, revealing dendrite colonies growing epitaxially through the layers from the substrate without a clear preferred growth direction. The average spacing between primary dendrite arms is 25.0 ± 10.0 µm.

From the macrostructural images of the horizontal cross-section, areas were selected for microstructural analysis (Figure 5e–g). From these cross-sectional images, the width of elongated columnar grains can be assessed (Figure 5e), with an average size of 80 µm. In the current study, cracking occurred post-solidification in a copper section of the joint. The volume fraction of crack defects in the samples was less than 5%. The cracking at the C11000-Inconel 625 interface (Figure 5f) occurred due to the mismatch in thermal expansion between the two metals. Nonetheless, intergranular cracking in the copper component can be attributed to solidification cracking during the final stage of solidification (Figure 5e), driven by the immiscibility between Cu and Inconel 625 alloying elements like Fe and Cr, leading to the creation of Ni- and Cu-rich liquids with varying freezing ranges [52,53].

Cracking during fabrication is largely explained by the residual thermal stresses caused by large temperature gradients and rapid solidification, leading to shrinkage of the melt pool. The dendritic structure of the nickel component in the horizontal cross-section relative to the growth direction (Figure 5g) is visualized as chains of characteristic “Maltese crosses”.

Figure 6 presents metallographic images of Type B (Figure 6a–c) and Type C (Figure 6d–f) bimetallic specimens in longitudinal sections relative to the grown wall. Structural differences between copper and nickel components are clearly identifiable: light areas correspond to the nickel-based superalloy, and contrasting areas correspond to the pure copper. The microstructure of the nickel component in the bimetallic specimens of Type B (Figure 6c) and Type C (Figure 6e) is identical to that described above: a complex heterogeneous dendritic structure with dendrite colonies growing epitaxially through layers from the substrate without a predominant growth direction. The average spacing between primary dendrite arms is 40.0 ± 10.0 µm.

Unlike the nickel component, the copper component shows variation in the grain structure along the height of the sample, manifested by changes in structural element sizes. In the copper component, large columnar grains form with curved boundary geometry in the Type B composite sample (Figure 6a) and straight boundary geometry in the bimetallic specimen Type C (Figure 6f). The zigzag boundaries are primarily related to the heat input level: lower heat input leads to the formation of zigzag grain boundary structure, while higher heat input leads to the grain boundary structure consisting of large, straight, elongated grains. The melt pool direction in one layer is opposite to that in the next, causing grains to grow obliquely relative to one of the directions perpendicular to the surface. Growth continues into the next layer if the scanning direction remains the same; if the scanning direction reverses, growth realigns away from the maximum heating direction.

When depositing the first pure copper layers onto already deposited nickel layers (Figure 6a) in Type C specimens, areas up to 1.5 mm long form with equiaxed copper grains averaging 40 µm in size. Then, a region up to 30 mm long with anisotropic grain structure develops, indicative of directional solidification during additive growth, with an average grain size of 900 µm. As the number of printed pure copper layers increases, the growth of large curved grains occurs, with grain thickness remaining roughly constant at 15 mm. In the middle segment of the copper component, rounded pores averaging 0.15 mm in size and irregularly shaped pores averaging 0.2 mm are observed. It is important to note that the volumetric fraction of defects does not exceed 0.2%.

Grains in the copper component of the bimetallic specimen Type C (Figure 6f) are elongated with an average size of 350 µm. The grain growth direction slightly deviates from vertical due to temperature gradients during printing. Grain boundaries in the copper component of the bimetallic specimens of Type C are straighter and wider compared to the zigzagging narrower grain boundaries in the copper component of the bimetallic specimens of Type B. Differences in the thermal conditions during 3D printing of the bimetallic specimens of Type C caused the formation of large columnar grains in the copper region, with growth trajectories oriented at approximately 80° to the printing direction.

The interface from the nickel region to the pure copper region in the specimen is sharp, with no defects such as cracks or lack of fusion detected (Figure 6b,e). The thickness of the interface from the nickel part to the pure copper part of the bimetallic specimen Type B reaches 120–150 µm (Figure 6b). At the interface between dissimilar materials, the contour of the melt pool is clearly visible. Differences in the thermophysical properties of the pure copper and nickel-base superalloy cause deep remelting of the previously deposited pure copper layer during the deposition of the nickel layer. However, no interphase interaction occurs between the nickel and copper components near the interface.

In the bimetallic specimen of Type C, the thickness of the interface between the copper and nickel alloy regions is less than 10 µm (Figure 6e). The interface between dissimilar materials is curved, which is related to high multidirectional thermal gradients due to the differing thermal conductivities of the materials. During the deposition of the top copper layer, it is constrained by the significantly cooler underlying layer, causing compressive deformation. At elevated temperature, the yield strength of the top layer decreases, allowing it to deform plastically in compression. Cooling of the plastically compressed top layer leads to its shrinkage, which restores tensile stress and also causes a bending curvature relative to the printing direction. This induces tensile stress in the growth direction.

The results of X-ray structural analysis of the bimetallic specimens of the C11000-Inconel 625 system are shown in Figure 7. All regions in the bimetallic specimens of Types A, B, and C have a cubic crystal lattice. The lattice parameters in the copper region of bimetallic specimens are a, b, c = 3.608 Å. The lattice parameters for copper differ between the core copper region and the interface in the bimetallic specimen of Type A, measuring a, b, c = 3.608 Å at the core and a, b, c = 3.594 Å near the interface, respectively. Based on the calculation of crystal lattice microstrain (Δ*d*/*d*) for the (111), (200), (220), (311), (222), and (400) planes, the average strain value of 1.9 × 10^−3^ was found. This leads to the rough estimate of a residual stress value of 300 MPa.

The lattice parameters for nickel superalloy also differ between the core region and the interfacial region of the bimetallic specimen Type A, being a, b, c = 3.580 Å and a, b, c = 3.566 Å, respectively. For microstrain calculations (Δ*d*/*d*), the same crystal planes were used, with average values in the copper and nickel of 0.9 × 10^−3^ and 3.8 × 10^−3^, respectively. The average residual stresses in the copper and nickel regions were estimated as 99 MPa and 798 MPa, respectively.

In the bimetallic specimens of Type B, no differences in the lattice parameters were observed for the copper and nickel alloy between the bulk (core) and the interface. The lattice parameters for the interface, copper, and nickel regions were a, b, c = 3.608 Å, a, b, c = 3.608 Å, and a, b, c = 3.617 Å, respectively. For the calculation of crystal lattice microstrains (Δ*d*/*d*), the planes (111), (200), (220), (311), (222), and (400) were used, with average values in the interfacial transition region and copper and nickel regions of 3.4 × 10^−3^, 1.0 × 10^−3^, and 5.2 × 10^−3^, respectively. The average residual stresses at the interface and in the copper and nickel alloy regions were 561 MPa, 110 MPa, and 1092 MPa, respectively.

For the bimetallic specimens of Type C, the lattice parameters for the interface, copper, and nickel components were a, b, c = 3.602 Å, a, b, c = 3.608 Å, and a, b, c = 3.615 Å, respectively. The same planes were used for microstrain calculations, with average values for the interface and copper and nickel regions of 2.9 × 10^−3^, 1.7 × 10^−3^, and 3.1 × 10^−3^, respectively. The average residual stresses at the interface and copper and nickel regions were 479 MPa, 187 MPa, and 651 MPa, respectively.

On the basis of EDS mapping of the elemental content in the interfacial zone of the bimetallic specimens C11000-Inconel 625 of Type A, the interface can be divided into two main areas: that of copper-based solid solution, and that of nickel-based solid solution (Figure 8a). Structural analysis of the test pieces confirms the gradual change in the chemical composition. No formation of secondary layers, delamination, foreign impurities, or lack of fusion defects were observed at the microscopic scale at the interface between the two dissimilar components of the bimetallic specimen. To investigate the features of structure formation and interphase boundaries in the samples, transmission electron microscopy analysis was performed on foils cut from the gradient zone (Figure 8c).

Diffraction patterns confirm the presence of the following phases: Ni solid solution, Cu solid solution, and M_6_C carbide phase. M_6_C-type carbides are rarely detected by scanning electron microscopy. Carbides can provide strengthening either directly (through dispersion strengthening) or indirectly (by stabilizing grain boundaries) [54]. M_6_C-type carbides (containing nickel, molybdenum, chromium, niobium) form in alloys and contribute to the strengthening of the γ-phase. During crystallization, carbides form early (since they have a higher melting temperature than other phase constituents). Because these carbides are enriched in chromium, the surrounding melt becomes depleted in chromium. In this case, the Cr depletion is minimal. The shape of these carbides is irregular, appearing as formations with non-uniform morphology.

### 3.3. Mechanical Properties and Fractography of the Bimetallic Specimens C11000-Inconel 625 of Types A, B, and C

To determine the mechanical and service properties, monotonic tensile and fatigue tests were conducted, impact toughness was determined, and material microhardness was evaluated. The results of the quasistatic tensile tests show that the nickel part exhibits a higher value of the ultimate tensile strength (UTS), yield strength, and elongation to failure, with average values observed at the interface and the lowest parameters in the copper region. When deformed in the printing direction, the nickel part shows lower elongation to fracture but higher strength values. The properties of the interface and the copper region are at similar levels to those measured in the growth direction.

The principal result of fatigue testing was the number of cycles to failure. Fatigue tests of the bimetallic specimens of Type A showed that test pieces cut from the interface and in the nickel and copper regions did not fail in the elastic deformation regime after 1,000,000 cycles. Testing in the elastoplastic regime led to failure of interface samples after 50,647 cycles when tested in the growth direction, or 2.5 times more cycles when tested in the printing direction.

Nickel samples fractured under elastoplastic conditions after 99,321 cycles when tested in the growth direction and after 1.3 times more cycles in the printing direction. In fatigue tests of the bimetallic specimens of Type B under elastoplastic deformation with high stress values, failure occurred only in the copper component, while no failure was observed in the elastic regime. Similar results were observed in fatigue tests of the bimetallic specimens of Type C. Fatigue testing along the printing direction showed no failure in the elastic regime within the test base, while increasing stress during cycling could cause failure after a large number of cycles, both for test pieces oriented with the stress application direction along the growth direction and for test pieces oriented in the printing direction.

The average microhardness of the bimetallic specimen of Type A in the transverse cross-section at the central point, measured away from the substrate, is 3.94 GPa for the nickel part and 1.03 GPa for the copper part. In the horizontal cross-section at the central point, the average microhardness for the nickel and copper components is 3.63 GPa and 0.83 GPa, respectively. The average microhardness in the gradient zone is 2.75 GPa. For the bimetallic specimens of Type B, the average microhardness of the nickel and copper parts is 3.56 GPa and 0.68 GPa, respectively, with 2.36 GPa at the interface. For the bimetallic specimens of Type C, average microhardness values are 2.77 GPa for the nickel part and 0.74 GPa for the copper part. The interface has an average microhardness of 4.01 GPa. This may result from the rapid cooling of the first nickel alloy layers deposited on copper. At 0.01 mm from the interface, microhardness values of the nickel decrease to approximately 2.7–2.8 GPa.

Impact toughness of the bimetallic specimen of Type A in the printing direction is 39.0, 19.0, and 25.0 J/cm^2^ for the nickel alloy part, copper part, and the interface, respectively. In the growth direction, the average impact toughness values for the nickel alloy and copper regions and the interface are 44.5, 22.5, and 26.5 J/cm^2^, respectively. Overall, these values are close to those measured in the printing direction. Failure occurs uniformly without delamination of the interface. Impact toughness of the bimetallic specimens of Type B in the growth direction is 32.0 J/cm^2^ and 3.1 J/cm^2^ for the nickel alloy and copper parts, respectively. Samples cut from the interface exhibit impact toughness of 9.8 J/cm^2^ (growth direction). The average impact toughness of Type C specimens is 41.0 J/cm^2^ for the nickel alloy part and 18.0 J/cm^2^ for the copper part. At the interface, the average impact toughness was 17.5 J/cm^2^.

Plastic deformation in the local areas demonstrates similar fracture characteristics (Figure 9). Most samples exhibit relatively high strength and elongation at break. Interface properties are also slightly lower, which is related to the greater sensitivity of miniature samples to local material properties in the test region. Tensile tests in the growth direction of bimetallic specimens of Type A show that plastic deformation in the interface and in the nickel alloy and copper parts develops fairly uniformly.

In the tensile tests of Type B pieces along the printing direction, plastic deformation and fracture behavior were found to depend strongly on the region from which the coupons were extracted. Samples from the nickel region exhibit pronounced high ductility and strength. Interface samples show intermediate mechanical properties and fracture features. Samples from the copper zone have low strength values but high elongation after fracture. Type C specimens exhibit tensile strength characteristics close to those of Type B specimens.

An examination of the distinct fracture characteristics observed in low-cycle fatigue testing of Type A specimens (Figure 10) indicates that fracture generally initiates with the development of a fatigue fracture zone 1, succeeded by a quasi-static fracture Type B as the specimen’s cross-section diminishes due to fatigue crack propagation. Type A specimens are distinct due to the lack of delamination at boundary 3 within the fatigue fracture zone, whereas delamination in component 4 occurs in the quasi-static tension zone, caused by the creation of a “neck,” the constriction of both components during plastic deformation, and their varying ductility. In the fracture area at the boundary, an increase in molybdenum 3 can be observed, which enhances the strength characteristics of the gradient zone. Failure in the fatigue crack zone 1 is mainly characterized by a stream fracture 6 and, to some extent, a dimple fracture 5, along with the development of secondary cracks 7. The static fracture zone displays a fracture pattern characteristic of ductile materials, featuring a dimpled structure in both the nickel and copper sections of the specimen. During tests on Type B or C specimens, deformation spreads through the most vulnerable component—copper (Figure 11). In this scenario, fatigue crack 1 first develops, showcasing a distinctive stream fracture structure 4, along with pores 3 on the fracture surface that do not serve as stress concentrators, and nickel particles 6, embedded significantly deeply within the copper. When a fatigue crack forms, the usable cross-section of the specimen diminishes, leading to failure under the applied force due to quasi-static loading and cyclic stresses that go beyond the material’s ultimate strength. This zone 2 features a dimpled fracture pattern (5), indicative of viscous materials fracture under static tensile stress.

Impact toughness tests show a relatively standard viscous fracture surface pattern for Type A specimens (Figure 12) and three primary fracture regions for Type B and C specimens (Figure 13, Figure 14 and Figure 15). During the testing of Type A specimens, fracture develops primarily according to viscous fracture with dimples 1, accompanied by delamination at the interface between components 2 and the appearance of secondary cracks 3 (Figure 11). Regarding the fracture mechanism, the specimens in this instance resemble those examined for cyclic fracture. In this instance, as opposed to static fracture or fatigue crack formation, a complex array of copper and nickel alloy fragments forms on the fracture surface in the delamination area of the component during impact. Additionally, the fragments show evidence of both intragranular viscous dimple fracture and interfacial fracture.

Testing specimens of Types B and C with a V-shaped impact concentrator applied to the boundary area reveals three types of fractures. The initial one manifests as a fracture primarily along the nickel section of the boundary zone, characterized by the development of curved fracture fronts on the fracture surface (Figure 13). The arrangement of the fracture fronts mirrors the characteristic stress distribution in the specimen during impact. The fracture pattern primarily exhibits ductility with a dimpled structure, although there is also secondary cracking in the nickel and the occurrence of substantial secondary cracks. The emergence of minor secondary fractures in the nickel section aligns with the existence of copper in these regions, suggesting that cracking occurs at the interfaces of the copper and nickel phases within the gradient zone. Nonetheless, in spite of the fracture along the boundary zone resulting from the position of the concentrator, neither brittle fracture nor significant defects in this zone were observed, suggesting a sufficiently strong bond between the nickel alloy and copper.

The second kind of fracture in the gradient zone for specimens like this is illustrated by a viscous dimple fracture 1 shifted toward the copper section of the boundary (Figure 14). The fracture structure shows pores 2, typical of printed copper, nickel alloy particles 3, and additional cracks 4. Samples exhibiting this type of fracture show reduced impact toughness values resulting from the formation of a primary crack along the most vulnerable component.

The third category of fracture is particularly characteristic of samples with severe component mixing in the boundary area (Figure 15). In this instance, the formation of a primary crack takes place via a viscous fracture mechanism along copper, resulting in a dimple fracture area 1, while also involving an interphase fracture mechanism at the interface of copper and spherical nickel particles of differing sizes 2, 3. Consequently, fractures in specimens of this kind happen precisely within the structural gradient area and exhibit a mixed character.

These results indicate the formation of a complex anisotropic structure in bimetals of different sequences of component deposition, influenced by additive manufacturing conditions promoting heat removal toward the substrate. The formed structure of large primary grains or dendrites elongated in the heat flux direction results in material property variations along the printing direction, growth direction, and angles between them. Studies of the bimetallic specimens of Types A, B, and C have shown that the interface region does not tend to weaken the assembly compared to the weaker (copper) part alone. Fracture during tensile testing occurs by the cohesive mechanism in the weaker of the two materials. In certain instances, a mechanism of adhesive–cohesive failure is noted during testing for cyclic strength or impact toughness. The acquired data show significant values of gradient zone characteristics in tensile, fatigue, and impact toughness assessments (Table 3), linked to the favorable metallurgical compatibility of copper and nickel throughout WEBAM.

## 4. Conclusions

The present study demonstrates that wire-feed EBAM enables the fabrication of the bimetallic specimens composed of C11000 pure copper and Inconel 625 nickel-based superalloy. Three different sequences of component deposition—lateral (Type A), Ni-on-Cu (Type B), and Cu-on-Ni (Type C)—were systematically investigated to elucidate their influence on microstructural development, interfacial characteristics, and mechanical properties.

The EBAM additive manufacturing process inherently induces strong microstructural anisotropy due to directional solidification governed by heat extraction toward the substrate or cooler component. In the copper regions, large columnar grains form, with morphology being highly sensitive to the thermal history: curved grain boundaries in Type B specimens reflect complex melt pool dynamics and lower effective heat input. In contrast, straighter, more uniform grains in Type C specimens arise from the high thermal conductivity of the underlying copper substrate, which stabilizes epitaxial growth. In contrast, the Inconel 625 region consistently develops a dendritic microstructure, with primary dendrite arm spacings ranging from 25 to 40 µm, depending on local solidification conditions. Critically, high-resolution microscopy and EDS mapping confirmed sharp, metallurgically sound interfaces with minimal intermixing (<10 µm in Types A and C; up to 150 µm in Type B due to deeper remelting of copper during nickel alloy deposition). No intermetallic compounds, cracks, or lack-of-fusion defects were detected. X-ray diffraction analysis further verified the absence of deleterious phase transformations at the interface.

Mechanical testing corroborates the integrity of the interface: tensile and fatigue failures consistently initiate in the weaker copper component rather than at the interface (cohesive failure). Interfacial impact toughness values (9.8–26.5 J/cm^2^) often exceed those of pure copper, suggesting that the interface may act as a mechanical buffer that redistributes stress. Moreover, fatigue life in the elastoplastic regime is significantly extended when the loading axis aligns with the printing direction, underscoring the importance of build orientation in structural design.

The breakdown of samples during testing is intricate and varied, primarily featuring viscous, cohesive fracture, along with adhesive bonding fracture in specific regions at the interfaces of different components. The unique arrangement of the gradient zone, characterized by unevenly distributed spherical particles in certain Type B and C samples, results in a hybrid failure pattern. The more consistent gradient zone and advantageous load direction in Type A specimens result in mainly viscous failure in the components, although secondary delamination between the components does take place. The most encouraging outcome of specimen testing is the lack of delamination or cracking in the boundary zone during fatigue tests while the primary fatigue crack develops, enabling us to anticipate favorable performance attributes of the specimens under loads that do not surpass the design values for the relevant products.

## Figures and Tables

**Figure 1 materials-18-05450-f001:**
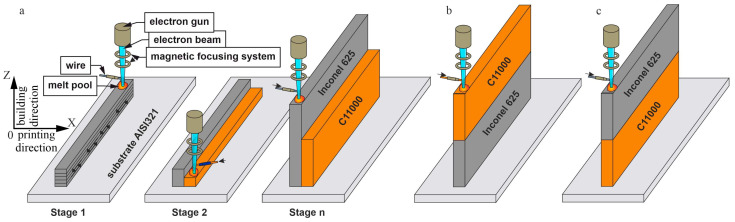
Strategy for printing bimetallic specimens from nickel-based superalloy and pure copper: (**a**) Type A—two single-hatch walls of nickel and pure copper; (**b**) Type B—lower part of wall made of nickel with pure copper deposited above; (**c**) Type C—lower wall made out of copper with nickel-based superalloy IN625 deposited above.

**Figure 2 materials-18-05450-f002:**
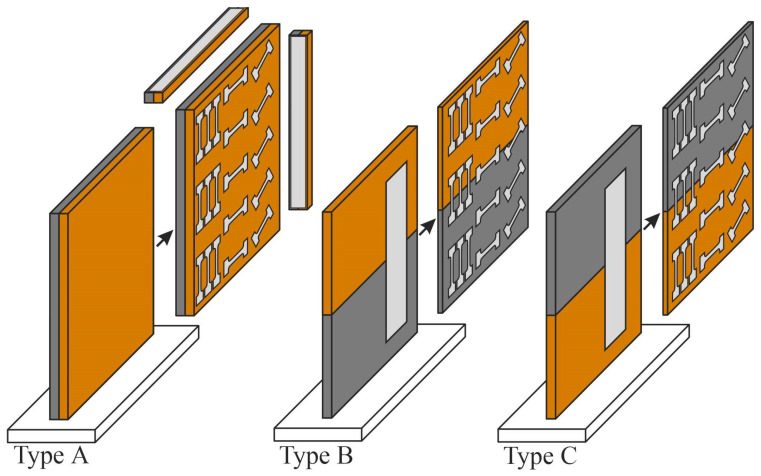
The cut-off scheme of bimetallic specimens of copper C11000-Inconel 625 of Types A, B, and C.

**Figure 3 materials-18-05450-f003:**
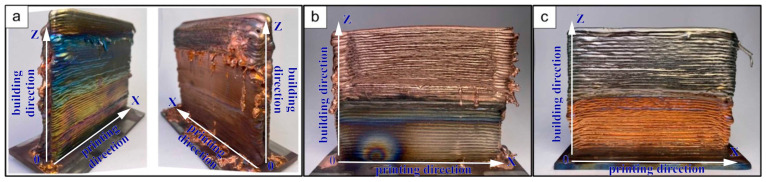
External appearance of the bimetallic specimens of the copper C11000-Inconel 625: (**a**) Type A—nickel and copper deposited side by side in parallel; (**b**) Type B—“Ni-on-Cu”; (**c**) Type C—“Cu-on-Ni”.

**Figure 4 materials-18-05450-f004:**
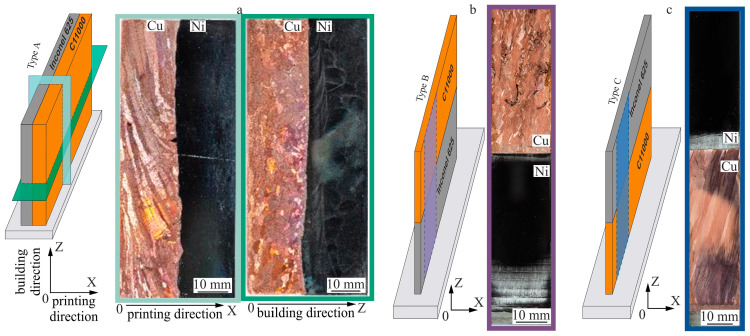
Macrostructure of the bimetallic specimens of copper C11000-Inconel 625 systems: (**a**) Type A—nickel and pure copper were co-deposited side by side in parallel; (**b**) Type B—Ni-on-Cu; (**c**) Type C—Cu-on-Ni.

**Figure 5 materials-18-05450-f005:**
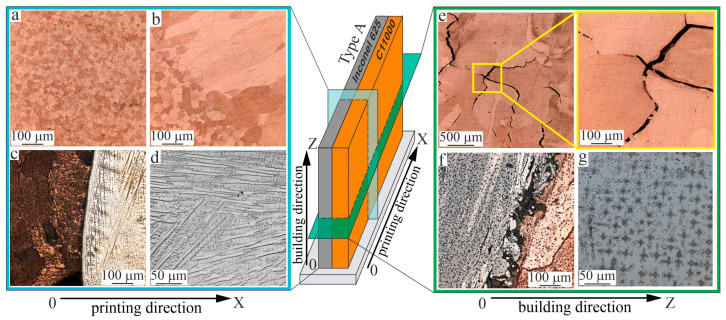
Microstructure of the bimetallic specimens of the copper C11000-Inconel 625 system, Type A: vertical section (**a**–**d**) and horizontal sections (**e**–**g**) in the lower segment of the test pieces.

**Figure 6 materials-18-05450-f006:**
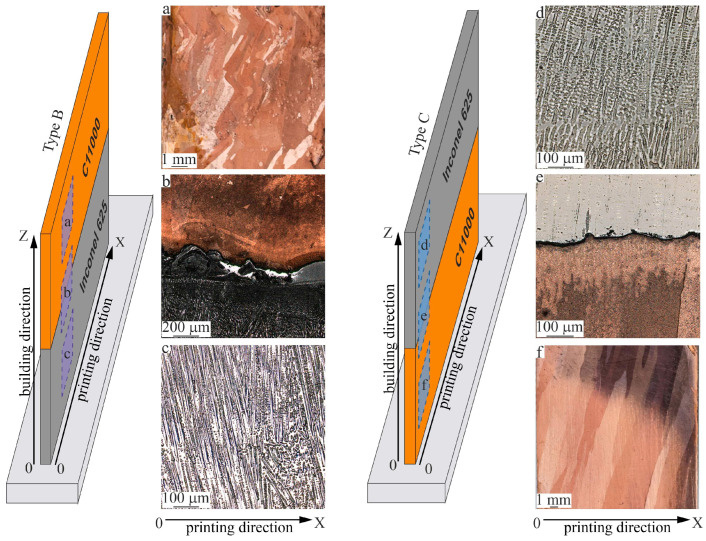
Microstructure of the bimetallic copper C11000-Inconel 625 specimens of Type B (**a**–**c**) and Type C (**d**–**f**) in the copper (**a**,**f**) and nickel (**c**,**d**) regions and near the interface (**b**,**e**).

**Figure 7 materials-18-05450-f007:**
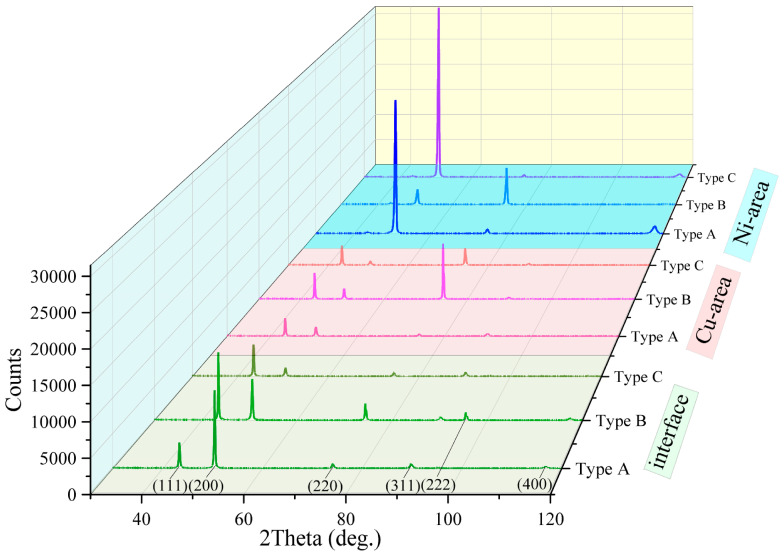
X-ray diffraction patterns of the bimetallic specimens of copper C11000-Inconel 625 of Types A, B, and C at the interface and in the copper and nickel regions.

**Figure 8 materials-18-05450-f008:**
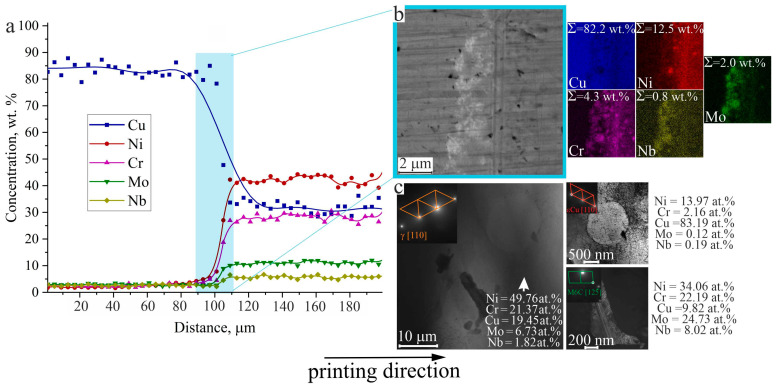
Analysis of the elemental composition by EDS mapping: (**a**) integrated line profiles, based on (**b**) maps of the area shown in the SEM image; (**c**) electron diffraction pattern obtained at the interface of the C11000-Inconel 625 system in the specimen of Type A.

**Figure 9 materials-18-05450-f009:**
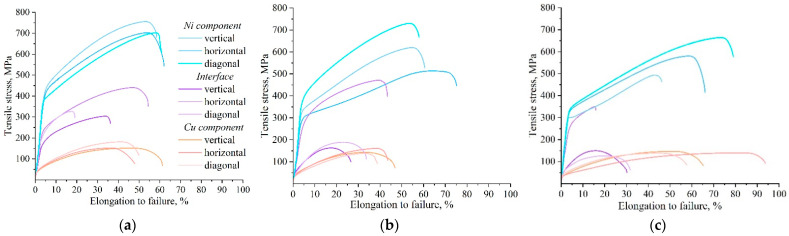
Representative tensile test diagrams for the test pieces made from bimetallic copper C11000-Inconel 625 specimens of Types A (**a**), B (**b**), and C (**c**).

**Figure 10 materials-18-05450-f010:**
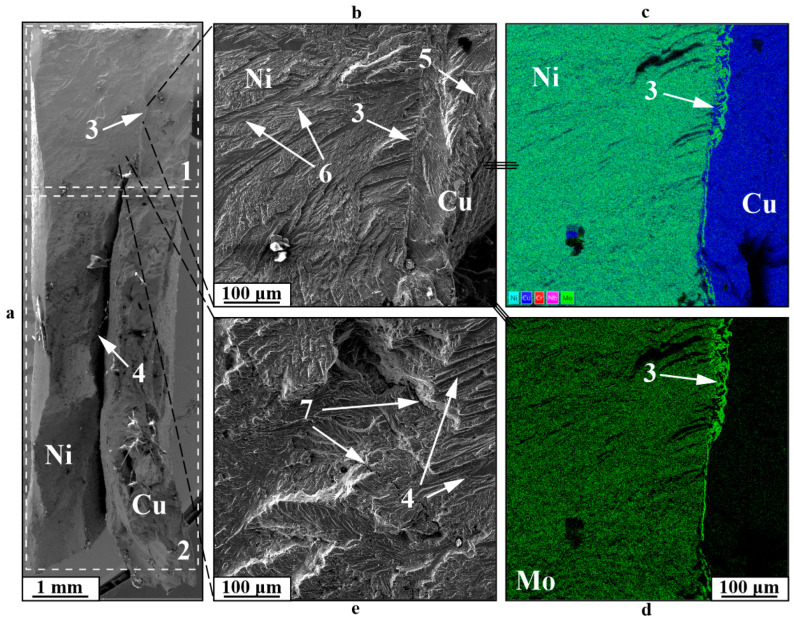
Fracture surfaces of a C11000-Inconel 625 bimetallic Type A specimen after fatigue testing: 1—fatigue fracture zone; 2—“static fracture” zone; 3—boundary between components; 4—boundary failure; 5—dimple fracture; 6—groove fatigue fracture; 7—secondary cracking; (**a**)—general view; (**b**,**e**)—enlarged images; (**c**,**d**)—EDS element distribution maps.

**Figure 11 materials-18-05450-f011:**
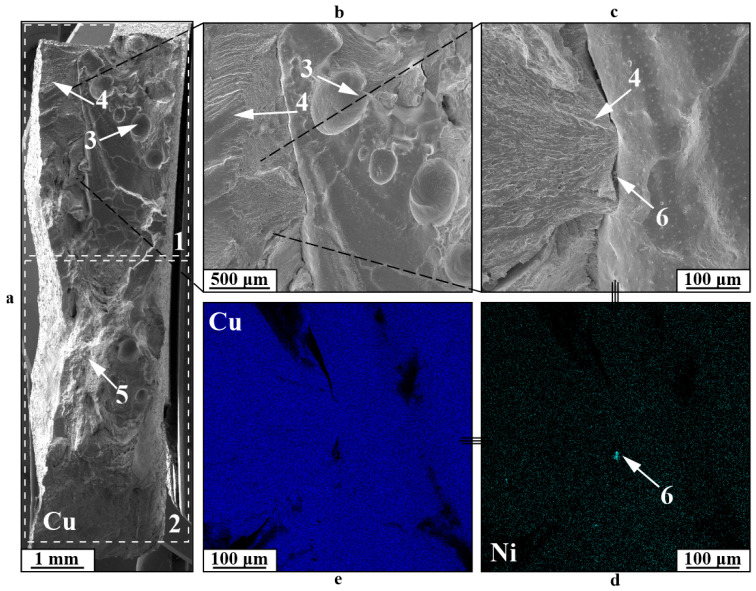
Fracture surfaces of a C11000-Inconel 625 Type B bimetallic specimen after fatigue testing: 1—fatigue fracture zone; 2—“static fracture” zone; 3—pores; 4—groove fatigue fracture region; 5—dimple fracture; 6—nickel particle; (**a**)—general view; (**b**,**c**)—enlarged images; (**d**,**e**)—EDS maps of element distribution.

**Figure 12 materials-18-05450-f012:**
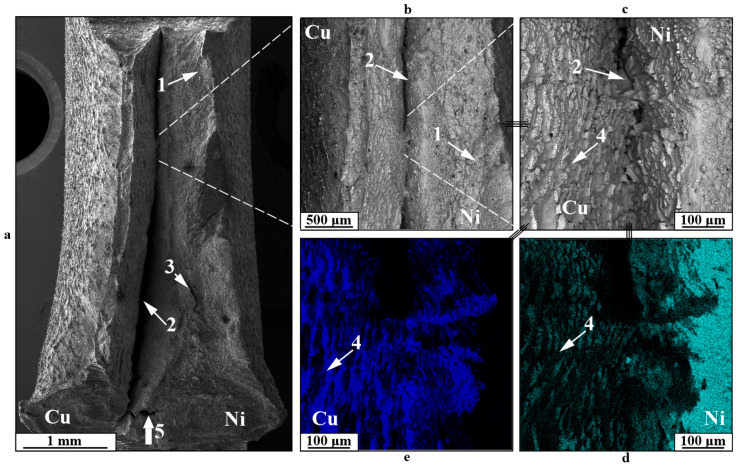
Fracture surfaces of a C11000-Inconel 625 Type A bimetallic specimen after impact testing: (**a**)—general view; (**b**,**c**)—enlarged images; (**d**,**e**)—chemical element distribution maps; 1—dimple fracture; 2—boundary delamination; 3—secondary cracks; 4—fragmented fracture; 5—impact location and direction.

**Figure 13 materials-18-05450-f013:**
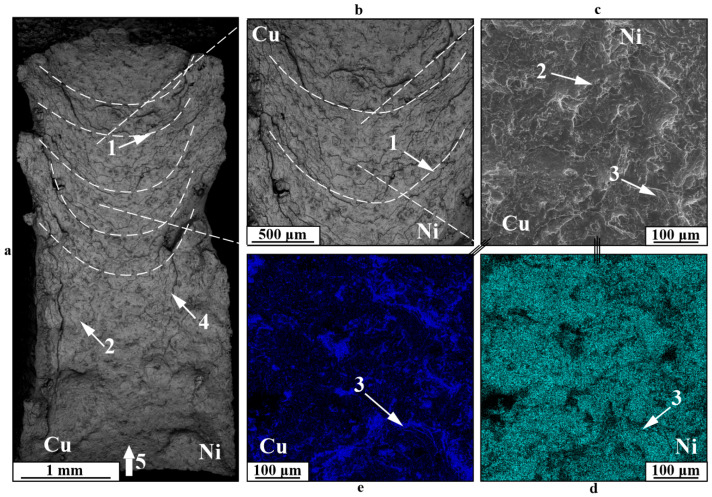
Fracture surfaces of a C11000-Inconel 625 Type B bimetallic specimen after impact testing: (**a**)—general view; (**b**,**c**)—enlarged images; (**d**,**e**)—EDS maps of distribution of chemical elements; 1—macroscopic traces of fracture; 2—dimple fracture area; 3—nickel cracking; 4—large secondary cracks; 5—location and direction of impact.

**Figure 14 materials-18-05450-f014:**
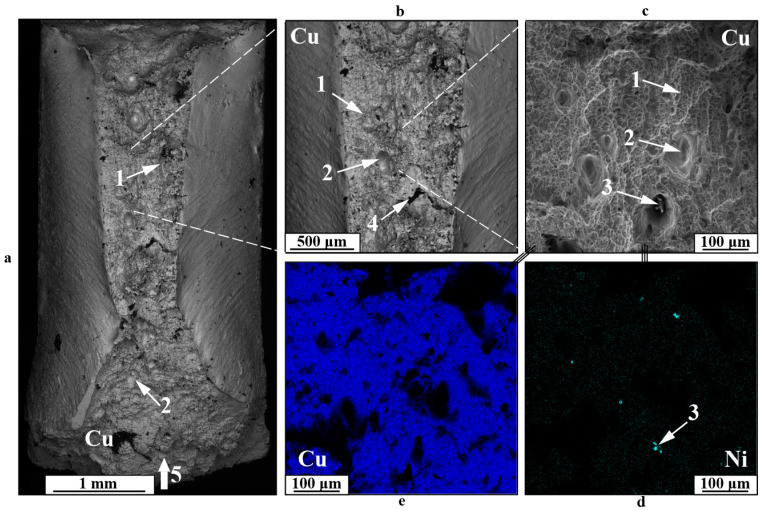
Fracture surfaces of a C11000-Inconel 625 Type C after bimetallic specimen after impact toughness testing: (**a**)—general view; (**b**,**c**)—enlarged images; (**d**,**e**)—EDS maps of distribution of chemical elements; 1—dimple fracture zone; 2—pores; 3—nickel particles; 4—secondary cracks; 5—location and direction of impact.

**Figure 15 materials-18-05450-f015:**
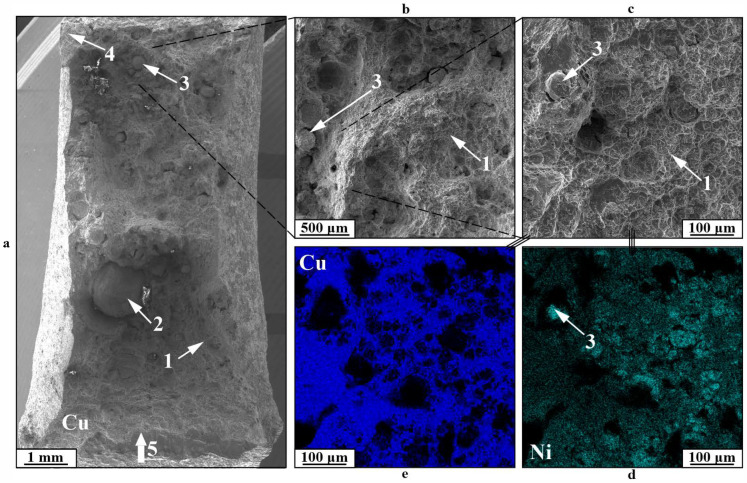
Fracture surfaces of a C11000-Inconel 625 Type C the bimetallic specimen after impact toughness testing: (**a**)—general view; (**b**,**c**)—enlarged images; (**d**,**e**)—maps of distribution of chemical elements 1—dimple fracture; 2—large spherical particles; 3—initiation of the main crack; 4—small spherical particles; 5—location and direction of impact.

**Table 1 materials-18-05450-t001:** Chemical composition of the feedstock used in the wire EBAM for the production of bimetallic specimens.

Material	Cu	Ni	Cr	Mo	Nb	Fe	Mn	C
C11000	99.9	up to 0.002	-	-	-	up to 0.005	-	-
Inconel 625	-	58.0	20.0–23.0	8.0–10.0	3.2–4.2	5.0	up to 0.05	up to 0.1

**Table 2 materials-18-05450-t002:** Technological process parameters of manufacturing bimetallic specimens using wire-feed EBAM.

Specimens	Material	Beam Current, mA	Scanning Speed, mm/min	Wire Feed Speed, mm/min	Estimated Layer Height, mm
Type A	Inconel 625	86.0 → 60.0	350.0	1392.6	1.0
	C11000	90.0 → 115.0	400.0	1591.5	1.0
Type B	Inconel 625	85.6 → 39.0	250.0	1293.1	1.3
	C11000	55.0	350.0	1392.6	1.0
Type C	Inconel 625	78.0 → 70.0	350.0	1392.6	1.0
	C11000	65.0	250.0	1293.1	1.3

**Table 3 materials-18-05450-t003:** Mechanical properties of samples with different types of structural gradient *.

Specimens	Impact Toughness, kJ/sm^2^	Cycles to Failure, ×1000	YS, MPa	UTS, MPa	Elongation, %
Type A	44.5	135 *	415	755	69
Type B	32.0	1000	350 *	730 *	72
Type C	41.0	1000	325 *	665 *	73

* Maximum gradient zone values.

## Data Availability

The original contributions presented in this study are included in the article. Further inquiries can be directed to the corresponding authors.

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
