# Peer review of "On the Heterogeneity of Deformation and Fracture in Bimetallic Specimens of the C11000-Inconel 625 System"

_materials, 2025, doi:10.3390/ma18235450_

Round 1

Reviewer 1 Report

Comments and Suggestions for Authors

In this study, bimetallic samples of copper C11000 Inconel 625 system were fabricated by multi wire electron beam additive technology, and three different deposition sequences were used to study.

I think this manuscript is very well written. Could be ranked top 10% of the papers in this journal Materials.

The working load is very heavy. The data are abundant. Many experimental methods are applied in this study.

Only a few comments from my side:

Introduction section, the research by WAAM or WEBAM on this inconel alloy and copper can be summarized. Is there any work on this topic?

Besides, how about their results?

For the materials, could the authors specify the precise chemical composition of the copper and inconel 625 alloys?

This wire raw material, where did the authors got it? For commercial producers or some else? And how about the processing history of those wires? This is important for the repeatability of the experiments.

Is the black parts in figure 4 e cracks or shrinkage?

In line 270 to 271, the solidification process and melt pool viscosity are discussed by references, however, the details of the melt viscosity of this study need to be implemented when discussing this topic.

Figure 6 is very good. However, the phases cannot be identified in this figure. It is better to mark the phases by legends.

Figure 7 can be divided into two figures, one figure for the line scanning and EDS, another figure for TEM result.

Some errors:

In abstract, the type I, II, and III are denoted. However, in the manuscript, type A, B, and C are named. Those should be revised to be consistent.

In addition, in line 305, type III need to be revised as type C.

Overall, this is a very good manuscript!

Author Response

Comments from the Reviewer 1:

In this study, bimetallic samples of copper C11000 Inconel 625 system were fabricated by multi wire electron beam additive technology, and three different deposition sequences were used to study.

I think this manuscript is very well written. Could be ranked top 10% of the papers in this journal Materials.

The working load is very heavy. The data are abundant. Many experimental methods are applied in this study.

Only a few comments from my side:

  1. Introduction section, the research by WAAM or WEBAM on this inconel alloy and copper can be summarized. Is there any work on this topic? Besides, how about their results?

A: Thank you. The Introduction section has been expanded to include an analysis of the current state of copper-nickel alloy research. The bibliography has been revised accordingly.

 For the materials, could the authors specify the precise chemical composition of the copper and inconel 625 alloys?

A: Thank you. In the section "Materials and conclusions" in Table 1 the chemical compositions of the wires is given.

 This wire raw material, where did the authors got it? For commercial producers or some else? And how about the processing history of those wires? This is important for the repeatability of the experiments.

A: The wire was custom-made by Metal Expedition. The wire was in as-delivered condition, and its chemical composition matched the specified grade.

 Is the black parts in figure 4 e cracks or shrinkage?

A: Thank you. Grain-boundary cracking results from high temperature gradients and solidification during fabrication. The value of the volume fraction of defects for samples does not exceed 5 %. Information added to lines 296-302.

 In line 270 to 271, the solidification process and melt pool viscosity are discussed by references, however, the details of the melt viscosity of this study need to be implemented when discussing this topic.

A: Thank you. Revised.

  1. Figure 6 is very good. However, the phases cannot be identified in this figure. It is better to mark the phases by legends.

A: Thank you. Corrected.

  1. Figure 7 can be divided into two figures, one figure for the line scanning and EDS, another figure for TEM result.

A: Thank you but authors would prefer to leave the results of microstructural analysis by scanning and transmission electron microscopy methods in Figure 7 without separation.

Some errors:

  1. In abstract, the type I, II, and III are denoted. However, in the manuscript, type A, B, and C are named. Those should be revised to be consistent.

A: Thank you. Corrected

 In addition, in line 305, type III need to be revised as type C.

A: Thank you. Corrected

Reviewer 2 Report

Comments and Suggestions for Authors

The manuscript is thematically suitable for the Materials journal. The results are presented in a logical and transparent way, the conclusions are correct. However, the following corrections should be made to the manuscript:

1. In the introduction, the authors should clearly identify the specific gap in the field that their work addresses.

2. The introduction requires supplementation with additional literature, which should be sourced and included.

3. The discussion of the results should be in a separate section or in a section together with the conclusions.

Author Response

Comments from the Reviewer 2:

The manuscript is thematically suitable for the Materials journal. The results are presented in a logical and transparent way, the conclusions are correct. However, the following corrections should be made to the manuscript:

  1. In the introduction, the authors should clearly identify the specific gap in the field that their work addresses.

A: Thank you. The Introduction section has been expanded to include an analysis of the current state of copper-nickel alloy research.

  1. The introduction requires supplementation with additional literature, which should be sourced and included.

A: Thank you. bibliography has been revised accordingly.

  1. The discussion of the results should be in a separate section or in a section together with the conclusions.

A: Thank you. However, we find it more convenient and readable to discuss the results directly as they are outlined. Conclusions is a section that summarizes al the results obtained and therefore should be a separate one. 

Reviewer 3 Report

Comments and Suggestions for Authors

The manuscript presents a solid and well-organized study on deformation and fracture behavior in C11000–Inconel 625 bimetallic specimens produced by wire-feed EBAM. The microstructural analysis, mechanical testing, and interface investigation are convincing. The manuscript is relevant for the additive manufacturing community. A key result is that the interface between copper and Inconel 625 is not a weak zone!!! All tensile and fatigue failures start in the softer copper, while the interface remains intact, even under cyclic loading. This shows that EBAM can produce strong and reliable Cu–Ni interfaces with minimal mixing and without harmful phases.

I recommend the article for publication after the authors address the following small points:

1. The explanation links grain shape to heat flow and melt pool behavior. Could the authors provide some quantitative thermal data or simple calculations to support this? For example, would a Kolmogorov model be suitable here?

2. Fatigue performance improves when loading and printing direction are aligned. Do the authors recommend specific build orientations for real components? Could the structure be arranged similar to architectured materials (see. works Y.Estrin, etc.), where geometry also improves performance?

3. Do the authors expect similar interface behavior in systems with larger differences in melting point or thermal conductivity?

Author Response

Comments from the Reviewer 3:

The manuscript presents a solid and well-organized study on deformation and fracture behavior in C11000–Inconel 625 bimetallic specimens produced by wire-feed EBAM. The microstructural analysis, mechanical testing, and interface investigation are convincing. The manuscript is relevant for the additive manufacturing community. A key result is that the interface between copper and Inconel 625 is not a weak zone!!! All tensile and fatigue failures start in the softer copper, while the interface remains intact, even under cyclic loading. This shows that EBAM can produce strong and reliable Cu–Ni interfaces with minimal mixing and without harmful phases.

I recommend the article for publication after the authors address the following small points:

  1. The explanation links grain shape to heat flow and melt pool behavior. Could the authors provide some quantitative thermal data or simple calculations to support this? For example, would a Kolmogorov model be suitable here?

A: Thank you. The formation of columnar grains is a commonly known phenomena during slow solidification in a melted state, including a melted pool formed by a welding arc or an electron beam. Often these columnar grains contribute to enhanced anisotropy of mechanical characteristics and therefore should be replaced for equiaxed ones. 

These grains always grow starting from the unmelted part of the pool, i.e. it is epitaxial growth mechanism, while the Johnson-Mel-Avrami-Kolmogorov (JMAK) model relates more to homogeneous sporadic nucleation in an infinite volume and is usually modelled using Phase Field, Molecular Dynamics and Cellular  Automata, etc., methods [https://doi.org/10.1016/j.addma.2020.101611]. Nevertheless, the JMAK model may be modified to describe the epitaxial  grain growth [https://doi.org/10.1103/5td9-y6w4]. However, we use additive manufacturing, i.e. layer-by-layer deposition, which means that depositing a layer over previously deposited one, always means that we have to remelt and reheat the underlying layers and thus cause structural and phase transformations in the ummelted part while epitaxial growth will restart in the remelted part. Taking all this into account including local thermal gradients would make the model too complicate and heavy.  there is all the more reason not doing so because columnar grains are deleterious and could be transformed into equiaxed ones using several known techniques. 

  1. Fatigue performance improves when loading and printing direction are aligned. Do the authors recommend specific build orientations for real components? Could the structure be arranged similar to architectured materials (see. works Y.Estrin, etc.), where geometry also improves performance?

A: Thank you. Various Types of bimetallic specimens were fabricated and studied to provide future engineers with a choice of expected mechanical properties and failure patterns for bimetals, depending on the intended purpose and design of the product. The recommendation of specific orientations and specimen Types depends on the intended purpose of the product.

  1. Do the authors expect similar interface behavior in systems with larger differences in melting point or thermal conductivity?

A: Thank you. This team of authors previously studied bimetallic products based on stainless steel and copper with different interfaces and determined the structural formation characteristics related to mechanical properties.